# Antibodies to the Spike Protein Receptor-Binding Domain of SARS-CoV-2 at 4–13 Months after COVID-19

**DOI:** 10.3390/jcm11144053

**Published:** 2022-07-13

**Authors:** Evgeniia A. Kolosova, Olga N. Shaprova, Daniil V. Shanshin, Valentina S. Nesmeyanova, Iuliia A. Merkuleva, Svetlana V. Belenkaya, Anastasiya A. Isaeva, Artem O. Nikitin, Ekaterina A. Volosnikova, Yuliya A. Nikulina, Marina A. Nikonorova, Dmitry N. Shcherbakov, Svetlana A. Elchaninova

**Affiliations:** 1State Research Center of Virology and Biotechnology VECTOR, Rospotrebnadzor, 630559 Koltsovo, Russia; ngelya209@gmail.com (O.N.S.); dan6091154224@gmail.com (D.V.S.); nesmeyanova_vs@vector.nsc.ru (V.S.N.); j.a.merkulyeva@gmail.com (I.A.M.); belenkaya.sveta@gmail.com (S.V.B.); anastasya_isaeva_1993@mail.ru (A.A.I.); a.nikitin1@g.nsu.ru (A.O.N.); volosnikova_ea@vector.nsc.ru (E.A.V.); dnshcherbakov@gmail.com (D.N.S.); 2Russian-American Anti-Cancer Center, Altai State University, 656049 Barnaul, Russia; 3Department of Infectious Diseases, Altai State Medical University, 656038 Barnaul, Russia; julia_didenko91@mail.ru (Y.A.N.); ma.nikulina@mail.ru (M.A.N.); 4Department of Biochemistry and Clinical Laboratory Diagnostics, Altai State Medical University, 656038 Barnaul, Russia; saelch@mail.ru

**Keywords:** COVID-19, antibodies, receptor-binding domain, spike protein, SARS-CoV-2, age, pneumonia

## Abstract

Identification of factors behind the level and duration of persistence of the SARS-CoV-2 antibodies in the blood is assumed to set the direction for studying humoral immunity mechanisms against COVID-19, optimizing the strategy for vaccine use, antibody-based drugs, and epidemiological control of COVID-19. Objective: This study aimed to study the relationship between clinical and demographic characteristics and the level of IgG antibodies to the RBD of SARS-CoV-2 spike protein after COVID-19 in the long term. Residents of the Altai Region of Western Siberia of Russia, Caucasians, aged from 27 to 93 years (median 53.0 years), who recovered from COVID-19 between May 2020 and February 2021 (*n* = 44) took part in this prospective observational study. The titer of IgG antibodies to the RBD of SARS-CoV-2 spike protein was measured repeatedly in the blood at 4–13 months from the beginning of the clinical manifestation of COVID-19 via the method of enzyme-linked immunosorbent assay. The antibody titer positively correlated with age (*p* = 0.013) and COVID-19 pneumonia (*p* = 0.002) at 20–40 and 20–24 weeks from the onset of COVID-19 symptoms, respectively. Age was positively associated with antibody titer regardless of history of COVID-19 pneumonia (beta regression coefficient *p* = 0.009). The antibody titer decreased in 15 (34.1%) patients, increased in 10 (22.7%) patients, and did not change in 19 (43.2%) patients from the baseline to 48–49 weeks from the onset of COVID-19 symptoms, with seropositivity persisting in all patients. Age and COVID-19 pneumonia are possibly associated with higher IgG antibodies to the spike protein RBD of SARS-CoV-2 following COVID-19 in the long term. Divergent trends of anti-RBD IgG levels in adults illustrate inter-individual differences at 4–13 months from the onset of COVID-19 symptoms.

## 1. Introduction

Discovering the pattern of acquiring and maintaining immunity to the novel coronavirus infection COVID-19 has become the focus of researchers around the world. The adaptive immune response to severe acute respiratory syndrome-related coronavirus 2 (SARS-CoV-2) is generally recognized as one defined by both cell and humoral immune responses [1,2]. Any attempts to evaluate cell immune response to COVID-19 have not yet led to an evidence-based choice of indicators and criteria for immunity levels or to standardized tests acceptable for a routine clinical practice setting. Consequently, the level and prevalence of specific SARS-CoV-2 antibodies are currently used as the major specific immunity indicators for choosing vaccination strategy and epidemiologic control of COVID-19 [3,4].

It is assumed that one of the main measurable indicators of immunity level to COVID-19 could be the level of IgG antibodies to the spike protein (S-protein) receptor-binding domain (RBD) of SARS-CoV-2 [5]. Greater attention to this indicator is explained by the fact that RBD plays a key role in COVID-19 progression. Via the S1 subunit of its spike protein, SARS-CoV-2 virion binds to a transmembrane angiotensin-converting enzyme 2 (ACE2) on target cells, which is followed by the virion penetrating the cell with further cytopathic effect [5]. Antibodies to the RBD of SARS-CoV-2 spike protein are capable of inhibiting virion’s interaction with target cells and, therefore, preventing infection [6]. These functional features have defined the choice of the S-protein RBD as a priority target for COVID-19 vaccines, with antibodies to this epitope serving as the main component in medication for treating this infection [3,7].

IgG antibody level to the SARS-CoV-2 RBD measured with immunochemical methods, including a widely available enzyme-linked immunosorbent assay, is positively associated with the neutralizing action of these antibodies against the virus [6]. This points to the possible use of the blood concentration of IgG antibodies to the SARS-CoV-2 RBD as a mass screening tool for immunity to COVID-19.

There are yet no definitive answers to the question of how long IgG antibodies to the SARS-CoV-2 RBD persist in the blood and what are their change patterns for the mid- and long-term period after COVID-19 [8]. This is explained by the difference in antibody measuring methods and non-uniformity of clinical study designs, which hinders meta-analysis and systematic review of results, as well as by the scarcity of results coming from long-term longitudinal observational studies [8].

Some clinical studies have detected a correlation between anti-RBD IgG level and sex, age, and COVID-19 severity mostly in the acute infection phase [8,9]. However, this association has not been found in other studies [8,9].

It can be assumed that, over a period of several months after COVID-19, the level of IgG antibodies to the RBD of SARS-CoV-2 spike protein and its trends depend on the disease course of COVID-19, as well as on demographic factors. In order to test this hypothesis, we repeatedly measured the change in the titer of IgG antibodies to the RBD by using an original method in patients after COVID-19 at 4–13 months after the onset of SARS-CoV-2 infection.

## 2. Materials and Methods

### 2.1. Study Design and Ethics

This long-term longitudinal serological observational study included the residents of the Altai Region of Western Siberia, Russia, after a COVID-19 recovery and matching inclusion criteria.

Inclusion criteria were a medically documented COVID-19 diagnosis in the patient’s records, a positive and a negative SARS-CoV-2 RNA test at the beginning and at the end of an acute phase of COVID-19, respectively, and any level of IgG antibodies to the RBD detected by an original enzyme-linked immunosorbent assay during the first visit of the observation period.

Exclusion criteria were an undetectable level of IgG antibodies to the RBD confirmed by retesting, symptoms of an acute respiratory infection, and/or a positive SARS-CoV-2 RNA test manifesting during the study.

Participants’ visits were scheduled 2 or 3 months after the previous visit. During each visit, blood was drawn to measure the titer of IgG antibodies to the RBD. The first visit also included the additional gathering of personal medical records about the disease course of COVID-19 and the results of naso-oropharyngeal swabs for SARS-CoV-2 RNA. During each following visit, information was gathered about symptoms of an acute respiratory infection since the previous visit, medical documentation of the infection, and results of naso-oropharyngeal swabs for SARS-CoV-2 RNA if applicable.

The study was conducted in line with the Good Clinical Practice, the 1975 Declaration of Helsinki amended in 2013, and the current Russian legislation on clinical research. The study protocol was approved by the Ethics Committee of the Altai State Medical University of Barnaul, Altai Region, Russia (Protocol 9 of 23 October 2020). All participants signed an informed consent form prior to being included in the study.

### 2.2. Methods for Obtaining RBD Recombinant Protein and Antibody Titer Measurement

For the enzyme-linked immunosorbent assay, an RBD recombinant protein corresponding to the S-protein RBD of SARS-CoV-2 (from V308 to N542) was used. To obtain it, a sequence coding the fragment of the spike protein RBD of SARS-CoV-2 was used (Wuhan-Hu-1, GenBank: MN908947). GeneOptimizer tool was applied to optimize DNA codon sequence for expression in CHO cells (https://www.thermofisher.com/ru/en/home/life-science/cloning/gene-synthesis/geneart-gene-synthesis/geneoptimizer.html, accessed on 1 July 2022). The resulting nucleotide sequence was synthesized in a pGH plasmid vector (OOO DNK-Sintez, Moscow, Russia). A pVEAL2-RBD integrative plasmid vector, coding the 308V-542N domain of the S-protein, was obtained based on the pVEAL2 vector. In the N-terminus, RBD contained the signal sequence of a tissue plasminogen activator (MDAMKRGLCCVLLLCGAVFVSA). In the C-terminus, the protein contained the 6× His sequence. The RBD-producing culture was obtained based on the Chinese hamster ovary CHO-K1 cell line. The cell transfection was performed with the pVEAL2-RBD plasmid using Lipofectamine 3000 (ThermoFisher, Waltham, MA, USA), following the producer’s manual. In order to integrate the vector expression cassette into the cell genome, the target plasmids were accompanied by added pCMV(CAT)T7-SB100 plasmid coding the SB100 transposase. Three days later, with a pVEAL2 vector containing a resistance gene to puromycin, this selective antibiotic (InvivoGen, San Diego, CA, USA) was added to the culture medium at 10 mcg/mL final concentration. The resistant clones were selected for three days, and then the polyclonal cell culture was transferred to a 96-well plate with one cell per well. Two weeks later, the wells were checked for monocolonies, the culture medium was selected, and clone yield was evaluated. The clones with the highest yield were cultivated in roll tubes, and the culture medium was collected.

The target protein was detected in the CHO-K1 culture medium via electrophoresis in 15% polyacrylamide gel in denaturing conditions and visualized via Coomassie blue staining and Western blotting using polyHis antibodies. The RBD recombinant protein was separated from the culture medium. For that, it was centrifugated to remove cell debris and filtered through 0.22 µm filtration systems. RBD was further purified through metal chelate affinity chromatography with Ni-IMAC Sepharose sorbent (GE Healthcare, Chicago, IL, USA). The RBD protein-to-column binding occurred at a flow rate of 1.5 mL/min. The unbound proteins were removed from the column with 5 volumes of washing buffer (40 mM imidazole, 30 mM NaH_2_PO_4_, 500 mM NaCl, pH 7.4) at a flow rate of 1.5 mL/min. RBD was eluted by 3 volumes of elution buffer (500 mM imidazole, 30 mM NaH_2_PO_4_, 500 mM NaCl, pH 7.4) at a flow rate of 1 mL/min. The next stage involved purification via ion-exchange chromatography on a series of columns with a cation exchange sorbent (SP-Sepharose) and an anion exchange sorbent (Q-Sepharose) balanced out by using 50 mM NaHCO_3_, pH 7.6. After loading the protein, the columns were washed with 50 mM NaHCO_3_, pH 7.6. Target protein fractions with optical density higher than 0.25 OD were analyzed via electrophoresis in denaturing conditions in 15% polyacrylamide gel. Protein impurities bound to Q- and SP-Sepharose sorbents were eluted by using a NaCl linear gradient 0 to 1 M concentration in 50 mM NaHCO_3_, pH 7.6.

RBD-containing fractions were analyzed via electrophoresis in denaturing conditions in 15% polyacrylamide gel, dialyzed against sodium-phosphate-buffered saline, and further subject to sterilization filtration through 0.22 µm filters. Quantitative analysis of the protein contained was performed using the Lowry method [10].

IgG antibodies to the RBD of SARS-CoV-2 spike protein were detected in the venous blood serum with an original enzyme-linked immunosorbent assay. Recombinant RBD was sorbed on a high-sorption capacity 96-well plate with 200 ng/well in sodium-phosphate buffer at 4 °C throughout the night. After 3 washes in an ELx50 plate washer (BioTek, Winooski, VT, USA) with a washing solution containing 0.5% polysorbate 20 in the sodium-phosphate buffer, the plate received 150 µL/well of blocking solution, which contained 1% powdered milk (Sigma-Aldrich, St. Louis, MI, USA) in sodium-phosphate buffer, and was incubated for 1 h at 37 °C in a PST-60HL plate thermo-shaker (Biosan, Rīga, Latvia) at 200 rpm. Later, 100 µL of blood serum in a dilution range of 1:100 to 1:3200 in blocking buffer solution was applied and incubated for 1 h at 37 °C in a PST-60HL plate thermo-shaker (Biosan, Rīga, Latvia) at 200 rpm. After 3 washes, the plate received 100 µL of goat antibody conjugate to the constant region of a human IgG antibody tagged with horseradish peroxidase (Sigma-Aldrich, St. Louis, MI, USA) in a working dilution of 1:20,000 and was incubated for 1 h at 37 °C in a plate thermo-shaker at 200 rpm. After 6 washes, the plate received 50 µL of a tetramethylbenzidine-based substrate and was incubated at room temperature for 20 min. The reaction was arrested by adding 50 µL of 1 M sulfuric acid. Optic density (absorbance) was measured at 450 nm wavelength on an iMark plate absorbance reader (Bio-Rad, Hercules, CA, USA).

The measurement result was considered if the arithmetic mean value of optical density (OD) did not exceed 0.15, with a positive control not lower than 1.0.

The recombinant RBD, obtained with the original enzyme-linked immunosorbent assay to IgG antibodies to the S-RBD of SARS-CoV-2, was further used to conduct research on 120 blood serum samples of healthy donors. The samples were collected in the Siberian Federal District, Russia, in October and November of 2019 and transported to the State Scientific Center of Virology and Biotechnology “Vector” (Koltsovo, Russia), as previously described [11]. IgG antibodies to the spike protein RBD of SARS-CoV-2 were detected in none of the samples, which corresponds to 100% specificity. The 1.0 µg/mL solution of the NB6 monoclonal antibody to the RBD of SARS-CoV-2 spike protein was used as a positive control for the enzyme-linked immunosorbent assay, obtained as presented by Schoof et al. (2020) [12]. Coefficient of variation (CV) for within-run reproducibility and between-run precision did not exceed 9.0% and 13%, respectively, within 0.1 and 3.0 OD.

Evaluation of the test precision was particularly difficult because we were unable to find a generally accepted gold standard—a commercial seroconversion panel with public information on the SARS-CoV-2 genome variant, which infected the serum panel donor, as well as information on the antigen coding sequences in the test systems for the panel validation. We have not yet planned to evaluate the precision of our test using data from the Russian regulatory authorities. However, we analyzed 314 serum samples of the Altai Region residents who recovered from COVID-19 during the same period as the research participants. Out of these serum samples, collected 1–14 months after COVID-19 onset, 273 samples (86.9%) were reactive to the antigen used. This allowed us to consider the precision of our test to be no less than 86%. With these values of test sensitivity and specificity, the calculated negative predictive values (NPV) were 74.5%, and the positive predictive values (PPVs), also called precision, were at 100%.

The 1:3200 titer limit was defined at the study planning stage since it seemed sufficient for achieving the objective of the study, considering previously conducted research studies. These studies have determined that IgG antibodies to the spike protein RBD of SARS-CoV-2 in the titer range of 1:400 to 1:3200, measured using the original enzyme-linked immunosorbent assay explored above, show a close correlation with effective virus-neutralizing protection of the Vero cells against the cytopathic effect of SARS-CoV-2 [13].

### 2.3. SARS-CoV-2 RNA Identification Methods

According to the medical records of the study participants, SARS-CoV-2 RNA identification in the mucous naso-oropharyngeal material at the beginning and end of the acute COVID-19 phase was performed in various certified clinical laboratories of the Altai Region via a real-time reverse transcription-polymerase chain reaction using chemical solutions approved for clinical use in the Russian Federation.

### 2.4. Statistical Analysis Methods

Statistical analysis was performed using Statistica 8.0 program (StatSoft. Inc, Tulsa, OK, USA). Non-parametric criteria were used in data analysis. The two-sided Fisher’s exact test and the Pearson’s chi-squared test (χ^2^) were used for the evaluation of intergroup differences in categorical variables, and the Mann–Whitney U test was used for numeric variables of two independent samples; in addition, the Kruskal–Wallis H test was for three and more independent samples, whereas the Friedman’s ANOVA with the Wilcoxon matched-pair test was used for three and more dependent samples. Relations between variables were evaluated using the Spearman correlation coefficient (r), as well as the multiple regression analysis. The odds ratio was calculated using logistic regression analysis. The significance level of *p* < 0.05 was set for all statistical criteria applied. The results for numeric variables are presented as sample arithmetic mean and standard deviation, with a 95% confidence interval, while the results for ordinal variables are presented as a median with an indication of an upper and a lower quartile or a minimum and a maximum.

## 3. Results

### 3.1. Participant Characteristics

The residents of the Altai Region who recovered from COVID-19 between May 2020 and February 2021 took part in our study. This time period was marked by two increases in the incidence of COVID-19 [14]. Study enrollment and collection of venous blood samples occurred from 11 February to 28 June 2021.

There were 224 subjects (180 women and 44 men, 95 of whom had COVID-19 pneumonia in medical history) aged 46.3 ± 13.3 years included in the study. However, 26 subjects (20 women and 6 men, 6 subjects had COVID-19 pneumonia in medical history) aged 52.0 ± 13.1 had no IgG antibodies to the RBD of SARS-CoV-2 detected at Visit 1 and were, therefore, excluded from the study.

Out of 198 subjects who continued to participate in the study, 140 subjects (113 women and 27 men) with a detectable level of antibodies discontinued their participation after Visit 1 or Visit 2 due to their wish to get vaccinated amid the continuing pandemic.

In total, 58 subjects (47 women and 11 men, 29 subjects had COVID-19 pneumonia in medical history) finished their participation in the study; however, 14 subjects (11 women and 3 men, 6 of whom had COVID-19 pneumonia in medical history) missed 1 of the 4 visits foreseen by the study protocol. Only 44 subjects made no fewer than 3 visits and, therefore, provided no fewer than three measurements of antibody titer in the planned observation period. There were no cases of patient premature discontinuation due to a repeat COVID-19 infection or an acute respiratory disease of a different etiology.

Statistical analysis was performed for a total of 44 patients, Caucasians (36 females assigned at birth and 8 males assigned at birth), with no less than 3 measurements of IgG antibodies to the RBD conducted throughout the study. The mean age of study participants stood at 52.5 ± 13.2 years (27 to 93 years, median at 53.0 years).

About 98% of 44 subjects had experienced mild and moderate cases of COVID-19, with the severity verified according to the existing at the time of patients’ infection Temporary Guidelines of the Ministry of Healthcare of the Russian Federation, Ed. 6–10 [15]. Some patients had a documented COVID-19 pneumonia complication confirmed by high-probability virus-induced changes in the lung tissue according to computer tomography scans (Table 1).

### 3.2. Titer of IgG antibodies to the RBD of SARS-CoV-2 Spike Protein over Time since the Onset of COVID-19

Repeated measuring of the titer of IgG antibodies to the RBD of SARS-CoV-2 spike protein was conducted between 16.1 and 51.0 weeks (4–13 months) since the manifestation of the first COVID-19 symptoms. All enrolled participants remained seropositive for IgG antibodies to the RBD of SARS-CoV-2 spike protein throughout the whole observation period, with the antibody titer ranging between 100 and 3200 in the study group (median 1600).

Through the analysis of all antibody titer measurements, it was established that the antibody titer did not correlate with the recentness of the first symptoms of COVID-19 (*n* = 148; r = 0.012; *p* = 0.888). This is coherent with no difference in the mean values of the antibody titer in the group, registered during the visits at different points in time since the onset of the infection (Table 2).

Evaluation of the antibody titer changes over time from the first to the last visit for each participant allowed us to break down all participants into three subgroups: 15 (34.1%) patients with lower antibody titer, 10 (22.7%) patients with no titer change, and 19 (43.2%) patients with higher antibody titer. Figure 1 shows trend charts for the antibody titer in these subgroups.

### 3.3. Correlation between IgG Antibodies to the RBD of SARS-CoV-2 Spike Protein and Demographic and Clinical Characteristics of Study Participants

Antibody titer, registered at visits differentiating in time elapsed since the onset of COVID-19, had no correlation with sex (Table 3).

At the first three visits, there was a positive correlation between the antibody titer and the patients’ age, while at the first visit, there was a positive correlation between COVID-19 pneumonia complication and the extent of lung damage according to CT scans (Table 3). At the same time, age of patients showed positive correlations with both pneumonia in the medical history (*n* = 44; r = 0.511; *p* < 0.001) and the extent of lung damage due to COVID-19 complication (r = 0.479; *p* = 0.001).

The strength of the relationship between the antibody titer and the discovered correlates, i.e., age and COVID-19 pneumonia, was evaluated with multiple linear regression analysis. The antibody titer at the first visit was chosen as a predictor variable since it is the visit when the correlation between the antibody titer and both age and COVID-19 pneumonia in the medical records of the patients was detected. Analysis results are shown in Table 4.

The value of the standardized beta regression coefficient was statistically significant only for age as a predictor variable of the antibody titer (Table 4). This points to a probability that age might influence the titer of IgG antibodies of the RBD of SARS-CoV-2 spike protein in the analyzed post-infection period independent of the viral COVID-19 lung damage. Judging by the value of the coefficient of determination R^2^ adjusted, about 27% of the antibody titer variability as a response variable was associated with age as a predictor variable.

The value and significance of the *F*-test for age after the adjustment for the presence of COVID-19 pneumonia in the medical history pointed to the signs of linear association between age and the antibody titer (Table 4). However, considering there are many factors behind the evolution of the post-infection immunity status that can vary in strength over time, a logistical regression analysis of the relationship between age and the anti-RBD IgG titer was performed discounting the factor of COVID-19 pneumonia in the medical history.

According to the results of logistical regression analysis, the probability of a median value of IgG antibodies to the SARS-CoV-2 RBD higher than 1600 at 4–13 months after the onset of COVID-19 symptoms was higher for the patients aged 55–93 than for patients aged 27–54 (odds ratio 5.7; confidence interval 1.5–21.6; *p* = 0.009).

### 3.4. Association of the Anti-RBD IgG Antibody Titer with the Demographic and Clinical Characteristics of the Study Participants, and the Initial Antibody Titer

The changes in the antibody titer from the first visit to the last did not correlate with the patients’ sex and negatively correlated with age, COVID-19 pneumonia complication, the extent of the lung damage according to CT scans, and the antibody titer at Visit 1 (Table 5).

Characteristics correlating with the changes in the anti-RBD IgG titer were analyzed for patient subgroups with different trends of the antibody titers throughout the observation period. It was established that the patient subgroups differed in age, the number of patients with COVID-19 pneumonia, and the antibody titer at Visit 1 (Table 6).

The subgroup with the antibody titer decreased by the end of the evaluated observation period was characterized as the oldest, with the highest presence of COVID-19 pneumonia in the medical records, and the highest antibody titer at Visit 1, compared with the patient subgroup with the antibody titer increase (Table 5).

## 4. Discussion

In this research, we evaluated the level of IgG antibodies to the SARS-CoV-2 RBD, which is viewed as the most clinically and epidemiologically significant component of humoral immune response to COVID-19 since the RBD of the spike protein plays a key role in the SARS-CoV-2 virions entering the cell [5,6].

The research results allowed us to assume that the IgG antibodies to the SARS-CoV-2 RBD may persist in the bloodstream for at least 13 months after infection. Comparison of this conclusion with the data from other studies is complicated due to their non-uniformity. As a recently conducted meta-analysis and a systematic review of 150 publications has shown [8], the results of most studies are based exclusively or primarily on the data from hospitalized patients and differ greatly in methodology, including the composition of participants, the length of observation, and antibody measurement methods. Very few studies included patients with mild and asymptomatic COVID-19 infections. Overall, though the antibody changes over time in the acute infection phase are described rather extensively, longer-term patterns are less apparent and do not allow to make a definitive conclusion [8]. Several studies have shown that, in most patients, IgG antibodies, including the ones to the SARS-CoV-2 RBD, are detected no less than one year after infection [16,17,18,19,20].

The established length of the seropositivity for IgG antibodies to the SARS-CoV-2 RBD is coherent with the data about the presence of RBD-specific memory B cells in more than 95% of patients recovered from COVID-19 [21]. The count of these cells is shown to remain relatively stable in the evaluated period of up to 12 months since the infection [22], and the level of antigen-binding IgG antibodies to the SARS-CoV-2 RBD expressed by these cells closely correlates with the virus-neutralizing capacity of antibodies [18,23]. One of the reasons why the signs of humoral immune response to SARS-CoV-2 persist for a year after COVID-19 infection is assumed to be the long presence of SARS-CoV-2 antigens, with the relevant immunoreactivity maintained particularly in the intestine [18].

The titer of IgG antibodies to the SARS-CoV-2 RBD varies greatly between individuals with the same amount of time elapsed since the onset of COVID-19. Moreover, patients after COVID-19 who have experienced nearly simultaneous surges in incidence rates in the same region might show divergent trends in the antibody titers for several months after the disease. This seems to demonstrate that, in order to determine tactics for possible recurrent infection management and indications for booster vaccination, it is reasonable to individually monitor humoral immunity to COVID-19.

Correlations established in the study allowed us to assume that age and the viral lung tissue damage, which largely determines the severity of the disease course, influence the changing pattern of the anti-RBD antibody titer in the mid and long term. These influences are most likely interconnected. Age is positively associated with a more severe disease course of COVID-19, pneumonia complications, and the extent of lung damage. As it follows from the collected data, the higher the age between 27 and 93 years and the greater the extent of lung damage during the disease, the more likely is a higher titer of IgG antibodies to the SARS-CoV-2 RBD at 4–5 months after the acute infection phase. On the other hand, the higher the antibody titer at 4–5 months after the disease, the more likely it is to decrease in the subsequent months.

Considering the results of the multiple regression analysis and comparison between the groups of participants with and without COVID-19 pneumonia complications, it can be assumed that independent of the virus-induced lung damage, an influence of age on the anti-RBD IgG antibody titer is very likely. We would like to note that almost simultaneously with our study establishing age and COVID-19 severity as factors in the anti-RBD IgG levels, several other publications appeared, also demonstrating a positive correlation between age and infection severity, and the level and virus-neutralizing capacity of antibodies to the spike protein [24,25,26,27,28].

At this time, the mechanism of how age influences the titer of IgG antibodies to the RBD of SARS-CoV-2 spike protein can only be hypothetically outlined. The pre-existing memory B cells to other HCoV coronaviruses are possibly capable of contributing to the formation of adaptive immunity to SARS-CoV-2 infection. Some studies point to such a possibility [29,30]; however, the efficiency of cross-protective immunity against different coronaviruses is generally understudied.

When discussing the cause-and-effect relationship between the antibody level and the severity of COVID-19, one cannot ignore the possibility of an antibody-dependent enhancement—an antibody-mediated binding of the virus with the Fc-receptor on immune-competent cells with the resulting enhancement of inflammatory response and greater severity of the infection course [25]. However, the data collected did not allow us to provide an evidence-based confirmation that the patients with relatively higher anti-RBD IgG levels after the disease had increased levels of these antibodies or antibodies to other epitopes of SARS-CoV-2 during the acute infection phase when the development of an antibody-dependent enhancement is indeed possible.

There are limitations to extrapolation to the population of the data on the persistence and the level of the IgG antibodies to the RBD of SARS-CoV-2 spike protein after COVID-19. These limitations are related to the wide inter-individual variability of the level of antibodies. The variability did not prevent us from establishing the presented facts with a generally accepted confidence threshold; nevertheless, it demonstrates that these facts require further confirmation by prospective cohort observation studies of post-COVID-19 immunity levels.

It should be highlighted that the persistence of the antibody-dependent protection against a recurrent SARS-CoV-2 infection requires additional study, considering that the immunochemical tests for specific antibodies do not fully reflect the virus-neutralizing capacity of antibodies. Another question that also remains open concerns the efficiency of antibody protective capacity against the new genetic lines of SARS-CoV-2 continuously appearing due to mutations. The SARS-CoV-2 mutations, especially in the regions coding the spike protein and its RBD, have been shown to lead to decreased efficiency in the antibody recognition of the virus if they were earlier induced by the virus of a different clade [31,32].

The study results point to the low probability of a relationship between the titer level of IgG antibodies to the RBD of SARS-CoV-2 spike protein and the biological sex. However, although in this study, the subject sample’s female-to-male ratio was 4.5, which might not be in line with the ratio in those who recovered from COVID-19 in the Altai Region, there has been no readily available information on this ratio. This defines the limitations on the application of our attained data regarding our assumption about the lack of relation between the level of IgG antibodies to the SARS-CoV-2 spike protein and the biological sex of those who recovered from COVID-19. We would like to highlight that, according to another systematic publication review [8], the results of the comparative analysis of the level of the antibodies to the RBD of SARS-CoV-2 spike protein in men and women have been inconclusive.

We believe it necessary to touch upon the matter of identifying the genetic variant of SARS-CoV-2 infecting the study participants since the virus mutations can possibly lead to some differences in the immune response to the infection. We did not sequence the SARS-CoV-2 extracted from the study participants. However, certain circumstances allowed us to claim with a high probability that the study participants were infected with the SARS-CoV-2 belonging to GR clade according to GISAID nomenclature (Pango Lineages A and B1.1.). First, all participants recovered from the disease from May 2020 to February 2021 when other genome variants of SARS-CoV-2 had not yet been registered as widespread in Russia and in the Altai Region, in particular [33]. Second, we used a recombinant RBD protein as the antigen in the enzyme-linked immunosorbent assay (https://www.multitran.com/m.exe?s=enzyme-linked+immunosorbent+assay&l1=1&l2=2, accessed on 1 July 2022) built on the basis of RBD SARS-CoV-2 sequence fragment, specifically of the GR clade (Wuhan-Hu-1, GenBank: MN908947). Therefore, the IgG antibodies to the RBD detected in all research participants who have recovered from COVID-19 for the first time may be viewed as specific to the mentioned genome variants of SARS-CoV-2, which circumstantially proves the etiological variant of the infection.

The presented study did not evaluate the influence of treatment on the level of antibodies to the RBD of SARS-CoV-2 spike protein. Such an evaluation was hindered because treatment strategies and the medication used differed greatly depending on the severity of the COVID-19 case but also differed for the patients with similar disease courses following the changes in the recommendations of the Ministry of Healthcare of Russia [15]. The question about the influence of treatment on the phenotype of the adaptive immune response to SARS-CoV-2 seems to require special research.

The results of the study led us to the conclusion that age and COVID-19 pneumonia are highly likely associated with a higher level of IgG antibodies to the RBD of SARS-CoV-2 spike protein in the first 4 to 6 months after the onset of COVID-19. In the subsequent period for up to almost a year, there was inter-individual variability in the changes over time of the anti-RBD IgG levels, with a probability of the antibody level remaining unchanged, decreasing, or increasing.

## Figures and Tables

**Figure 1 jcm-11-04053-f001:**
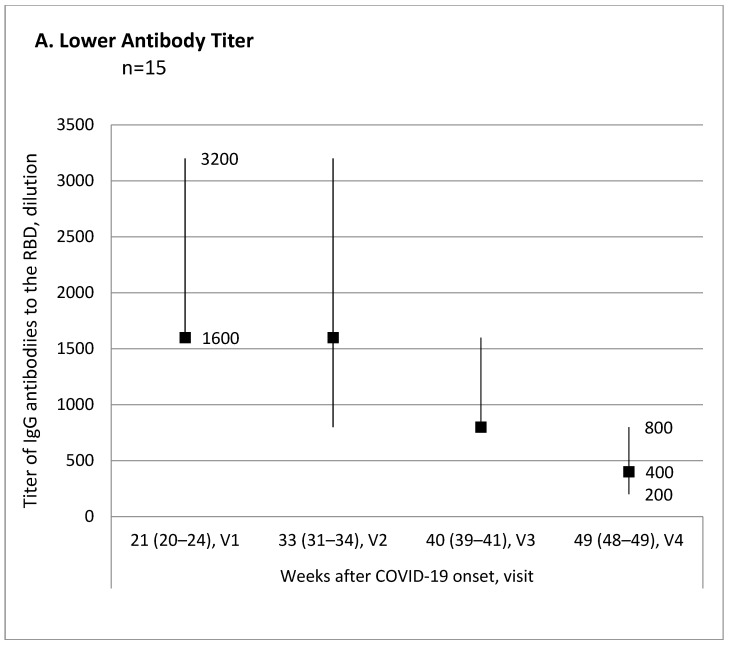
Titer trends over time of IgG antibodies to the RBD of SARS-CoV-2 spike protein at 4–13 months after COVID-19 onset. Antibody titer and the time since COVID-19 onset are shown in the median and interquartile range values. (**A**) Friedman ANOVA Chi-square = 16.7; *p* < 0.001. Wilcoxon Matched Pairs Test *p* values: 0.760 V1-V2; 0.069 V1-V3; 0.001 V1-V4; 0.043 V2-V3; 0.001 V2-V4; 0.028 V3-V4. (**B**) Friedman ANOVA Chi-square = 6.6; *p* = 0.084. (**C**) Friedman ANOVA Chi-square = 10.9; *p* = 0.012. Wilcoxon Matched Pairs Test p values: 0.002 V1-V2; <0.001 V1-V3; 0.018 V1-V4; 0.308 V2-V3; 0.068 V2-V4; 0.480 V3-V4.

**Table 1 jcm-11-04053-t001:** Participant medical history of COVID-19 (*n* = 44).

Disease Characteristics	Number of Patients, *n* (%)
SARS-CoV-2 RNA naso-oropharyngeal swabs conducted:	
Diagnostic test	44 (100.0)
Viral shedding test	44 (100.0)
SARS-CoV-2 RNA detected in the diagnostic test	44 (100.0)
No SARS-CoV-2 RNA detected in the viral shedding test	44 (100.0)
COVID-19 severity:	
Mild	21 (47.7)
Moderate	22 (50.0)
Severe	1 (2.3)
Critical	Not identified
COVID-19 pneumonia complication	23 (52.3)
Extent of high-probability virus-induced lung tissue changes according to CT scans of pneumonia patients:	
Minimal, up to 25%	20 (87.0)
Medium, 25–50%	3 (13.0)
Significant, 51–75%	Not identified
Subtotal, above 75%	Not identified

**Table 2 jcm-11-04053-t002:** Titer of IgG antibodies to the RBD of SARS-CoV-2 spike protein at 4–13 months after COVID-19 onset.

Visit	Number of Patients, *n*	Time between First COVID-19 Symptoms and Examination, Weeks		Antibody Titer, Dilution	
		Median (Interquartile Range)	Minimum–Maximum	Median (Interquartile Range)	Significance of Differences
1	44	21.8 (20.0–24.4)	16.1–26.0	800 (200–2400)	Friedman ANOVA Chi-square = 6.4; *p* = 0.095
2	41	32.2 (30.9–34.0)	26.6–35.3	1600 (800–3200)	
3	34	39.2 (37.9–40.0)	35.7–41.0	1600 (400–3200)	
4	29	48.0 (47.9–49.0)	41.0–51.0	800 (400–1600)	

**Table 3 jcm-11-04053-t003:** The Spearman correlation coefficients for demographic and clinical characteristics and the titer of IgG antibodies to the S-RBD of SARS-CoV-2 at 4–13 months after onset of COVID-19 symptoms.

Characteristic	Titer of IgG Antibodies to the RBD			
	Visit 1 Weeks 21.8 (20.0–24.4) *n* = 44	Visit 2 Weeks 32.2 (30.9–34.0) *n* = 41	Visit 3 Weeks 39.2 (37.9–40.0) *n* = 34	Visit 4 Weeks 48.0 (47.9–49.0) *n* = 29
Sex	−0.071; *p* = 0.648	−0.146; *p* = 0.362	−0.162; *p* = 0.359	−0.275; *p* = 0.148
Age	0.568; *p* < 0.001	0.327; *p* = 0.037	0.344; *p* = 0.047	−0.152; *p* = 0.430
Signs of viral pneumonia according to computer tomography scans	0.462; *p* = 0.002	0.271; *p* = 0.086	0.138; *p* = 0.435	−0.270; *p* = 0.157
Extent of high-probability virus-induced lung tissue changes according to CT scans	0.467; *p* = 0.001	0.215; *p* = 0.177	0.106; *p* = 0.551	−0.260; *p* = 0.174

The time since onset of COVID-19 symptoms is shown in the median and interquartile range values.

**Table 4 jcm-11-04053-t004:** Results of multiple linear regression analysis for relationship between the titer of IgG antibodies to the RBD of SARS-CoV-2 spike protein, age, and COVID-19 pneumonia.

Predictor Variables	Standardized Beta Regression Coefficient	Standard Error SE	95% Confidence Interval	*t*-Statistic	*p*-Value
Age	0.41	0.15	0.11–0.72	2.7	0.009
Signs of viral pneumonia on CT scans	0.21	0.15	−0.10–0.52	1.4	0.175

Predictor variable is the titer of IgG antibodies to the RBD of SARS-CoV-2 spike protein at Visit 1 at 21.8 (20.0–24.4) weeks after onset of COVID-19 symptoms. Analysis was conducted for all study participants (*n* = 44) with no variable values removed as an outlier. Coefficient of determination R^2^ multiple = 0.31. Coefficient of determination R^2^ adjusted = 0.27. *F*-test adjusted = 8.91; *p* = 0.001.

**Table 5 jcm-11-04053-t005:** The Spearman correlation coefficients for the changes in the titer of IgG antibodies to the RBD of the SARS-CoV-2 spike protein over the course of observation and demographic and clinical characteristics, and the antibody titer at Visit 1.

Characteristic	Spearman Correlation Coefficient; *p*-Value *n* = 44
Sex	−0.086; *p* = 0.577
Age	−0.428; *p* = 0.004
Signs of viral pneumonia according to CT scans	−0.391; *p* = 0.005
Extent of high-probability virus-induced changes in the lung tissue according to CT scans	−0.407; *p* = 0.006
Titer of IgG antibodies to the RBD of SARS-CoV-2 spike protein at Visit 1	−0.753; *p* < 0.001

**Table 6 jcm-11-04053-t006:** Comparison of age, COVID-19 pneumonia, and anti-RBD IgG antibody titer at Visit 1 in patient subgroups with divergent trends of the antibody titer at 4–13 months after onset of COVID-19 symptoms.

Characteristic	Antibody Titer Change			Difference Significance
	Subgroup 1	Subgroup 2	Subgroup 3	
	Decrease *n* = 15	Not Detected *n* = 10	Increase *n* = 19	
Age, years, median (interquartile range)	57.0 (53.0–63.0)	56.6 (51.0–60.0)	48.0 (38.0–55.0)	Kruskal–Wallis test H = 10.6 *p* = 0.005 Mann–Whitney U test: p_1–2_ = 1.0 p_1–3_ = 0.005 p_2–3_ = 0.190
Number of patients with signs of viral pneumonia according to CT scans, *n* (%)	13 (85.7%)	4 (40.0%)	6 (31.6%)	Pearson Chi-square: 11.0 *p* = 0.004 Fisher two-tailed: p_1–2_ = 0.028 p_1–3_ = 0.002 p_2–3_ = 1.0
Titer of IgG antibodies to the RBD of SARS-CoV-2 spike protein Visit 1	1600 (1600–3200)	1600 (400–3200)	200 (100–1600)	Kruskal–Wallis test H = 26.8; *p* < 0.001 Mann–Whitney U test: p_1–2_ = 1.0 p_1–3_ < 0.001 p_2–3_ = 0.001

All three subgroups did not differ in the time elapsed since onset of COVID-19 symptoms at Visit 1 (Kruskal–Wallis test H = 0.3; *p* = 0.823).

## Data Availability

Not applicable.

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
