# Peer review of "Antibodies to the Spike Protein Receptor-Binding Domain of SARS-CoV-2 at 4–13 Months after COVID-19"

_jcm, 2022, doi:10.3390/jcm11144053_

Round 1

Reviewer 1 Report

Kolosova et.al reported a relationship between age/clinical characteristics and the level of IgG antibodies to the RBD of SARS-CoV-2 spike protein in the first 4 to 6 months after the onset of COVID-19. I have two major concerns.

First of all, the sample size of this study (n = 44) was a bit small, the authors should increase the number of participants to verify this connection between age/clinical characteristics and the level of IgG antibodies to the RBD of SARS-CoV-2 spike protein.

Secondly, since S protein of Omicron is quite different than that of the other SARS-CoV-2 variants, I think the authors should consider distinguishing the variants in which the subjects were infected, as well as the specificity of IgG antibodies to the spike protein RBD of different SARS-CoV-2 variants.

Author Response

  1. First of all, the sample size of this study (n = 44) was a bit small, the authors should increase the number of participants to verify this connection between age/clinical characteristics and the level of IgG antibodies to the RBD of SARS-CoV-2 spike protein.

Response:

Thank you for noting this aspect. Unfortunately, it was impossible for us to recruit a greater number of the study participants than indicated in the manuscript.

It is due to the fact that many patients who had recovered from COVID-19 refused to take part in the research or discontinued their participation since they intended to get vaccinated in the planned research period of 4-12 months since the onset of the infection symptoms. Besides, in view of the possible spread of new genome variants of SARS-CoV-2 in the region, we considered it to be not entirely correct to include patients who had recovered from COVID-19 in the later periods of the pandemic in the region.

It is worth highlighting that despite the bit small sample size, the results obtained are in line with the generally accepted confidence threshold. The limitations of the some study conclusions are discussed in the manuscript.

  1. Secondly, since S protein of Omicron is quite different than that of the other SARS-CoV-2 variants, I think the authors should consider distinguishing the variants in which the subjects were infected, as well as the specificity of IgG antibodies to the spike protein RBD of different SARS-CoV-2 variants.

Response:

Thank you for pointing this out. We have reviewed the possible differentiation between the variants of SARS-CoV-2 infecting the subjects and the specificity of IgG antibodies to the RBD. We have added the discussion of the matter to the manuscript (see pp. 15) and we are quoting the abstract here.

Addition to the manuscript:

We believe it necessary to touch upon the matter of identifying the genetic variant of SARS‑CoV‑2 infecting the study participants since the virus mutations can possibly lead to some differences in the immune response to the infection. We have not sequenced the SARS-CoV-2 extracted from the study participants. However, certain circumstances allow us to say with high probability that the study participants were infected with the SARS-CoV-2 belonging to GR clade according to GISAID nomenclature (Pango Lineages A and B1.1.). First of all, all participants recovered from the disease from May 2020 to February 2021 when other genome variants of SARS-CoV-2 have not yet been registered as widespread in Russia and in the Altai Region, in particular [33]. Second, we used a recombinant RBD-protein as the antigen in the enzyme-linked immunosorbent assay built on the basis of RBD SARS-CoV-2 sequence fragment, specifically of the GR clade (Wuhan-Hu-1, GenBank: MN908947). Therefore, the IgG antibodies to the RBD detected in all research participants who have recovered from COVID-19 for the first time may be viewed as specific to the mentioned genome variants of SARS-CoV-2 which circumstantially proves the etiological variant of the infection.

In References

  1. Virus Genome Aggregator of Russia (VGARus). Available online: https://genome.crie.ru/app/inde/ (accessed on 01 Jule 2022)

Reviewer 2 Report

1.      In this study, the authors analyzed by their original antibody test, but it is better to describe the quality and accuracy of this test.

2.      The authors should show the number of people initially enrolled in the study. They should also describe the number of participants by gender at the time of study enrollment. And the number of people excluded by the exclusion criteria of this study and their characteristics should be shown.

 3.      Many of the people who actually participated in this study are women. Please consider whether that does not affect this study.

 4.      This study excludes participants who tested negative for antibodies at the time of study enrollment. Such individuals should also be included in the study to study changes in their antibody titers.

Author Response

  1. In this study, the authors analyzed by their original antibody test, but it is better to describe the quality and accuracy of this test.

Response:

Thank you for your attention to our antibody test. We have added a description of the test to the Methods section (see pp. 4), and we are quoting the abstract here.

Addition to the manuscript:

Evaluation of the test precision was particularly difficult because we were unable to find a generally accepted gold standard – a commercial seroconversion panel with public information on the SARS-CoV-2 genome variant which infected the serum panel donor, as well as information on the antigen coding sequences in the test systems for the panel validation. We have not yet planned to evaluate the precision of our test by the Russian regulatory authorities. However, we have analyzed 314 serum samples of the Altai Region residents who recovered from COVID-19 during the same period as the research participants. Out of these serum samples, collected 1-14 months after COVID-19 onset, 273 samples (86.9%) were reactive to the antigen used. This allows us to consider the precision of our test to be no less than 86%. With these values of test sensitivity and specificity, the calculated negative predictive values (NPV) were 74.5%, the positive predictive values (PPV), also called precision, were at 100%.

  1. The authors should show the number of people initially enrolled in the study. They should also describe the number of participants by gender at the time of study enrollment. And the number of people excluded by the exclusion criteria of this study and their characteristics should be shown.

Response:

Thank you for pointing that out. We have amended the text of the manuscript (see pp. 5), having added the following abstract.

There were 224 subjects (180 women, 44 men, among them 95 subjects with COVID-19 pneumonia in medical history) aged 46.3±13.3 years included in the study. But 26 subjects (20 women, 6 men, among the 6 subjects with COVID-19 pneumonia in medical history) aged 52.0±13.1 had no IgG antibodies to the RBD of SARS-CoV-2 detected at Visit 1 and were therefore excluded from the study.

Out of 198 subjects who continued to participate in the study, 140 subjects (113 women, 27 men) with a detectable level of antibodies discontinued their participation after Visit 1 or Visit 2 due to their wish to get vaccinated amid the continuing pandemic.

Fifty-eight subjects (47 women, 11 men, among them 29 subjects with COVID-19 pneumonia in medical history) finished their participation in the study, however 14 subjects (11 women, 3 men, among them 6 subjects with COVID-19 pneumonia in medical history) missed one out of four visits foreseen by the study protocol. Only 44 subjects made no fewer than three visits and therefore provided no fewer than three measurements of antibody titer in the planned observation period. There were no cases of patient premature discontinuation due to a repeat COVID-19 infection or an acute respiratory disease of a different etiology.

  1. Many of the people who actually participated in this study are women. Please consider whether that does not affect this study.

Response:

Thank you for the advice. We have changed the discussion of this matter in the manuscript (see pp. 15) as quoted here.

The study results point to the low probability of a relation between the titer level of IgG antibodies to the RBD of SARS-CoV-2 spike protein and the biological sex. However, while in the study subject sample the female-to-male ratio was 4.5, which might not be in line with the ratio in those who recovered from COVID-19 in the Altai region, there has been no readily available information on this ratio. This defines the limitations on the application of our attained data regarding our assumption about the lack of relation between the level of IgG antibodies to the SARS-CoV-2 spike protein and the biological sex of those who recovered from COVID-19. We would like to highlight that according to the systematic publication review [8], the results of the comparative analysis of the level of the antibodies to the RBD of SARS-CoV-2 spike protein in men and women have been inconclusive.

  1. This study excludes participants who tested negative for antibodies at the time of study enrollment. Such individuals should also be included in the study to study changes in their antibody titers.

Response:

Thank you for your attention to this part of the study.

Seronegative people were not included in the study according to one of the exclusion criteria. This criterion seems to us to be in line with the purpose of the study: to evaluate the dynamic of the level of IgG antibodies to the RBD of the SARS-CoV-2 spike protein after COVID-19 in the long term. We believe it to be unreasonable to continue observation of the subjects displaying a negative serum reaction beyond 4 months after the onset of COVID-19 symptoms due to the low probability of a later emergence of antibodies in a following period. 

Round 2

Reviewer 1 Report

The authors have addressed all issues raised by the reviewers.

Reviewer 2 Report

Thanks to the authors for responding appropriately to my comments.